# Multifaceted antibiotic stewardship intervention using a participatory-action-research approach to improve antibiotic prescribing for urinary tract infections in frail elderly (ImpresU): study protocol for a European qualitative study followed by a pragmatic cluster randomised controlled trial

Esther A R Hartman ,[1,2] Wim G Groen,[1] Silje Rebekka Heltveit-Olsen,[3] Morten Lindbaek,[3] Sigurd Hoye,[3] Pär-Daniel Sundvall,[4,5] Ronny Gunnarsson,[4,5] Ingmarie Skoglund,[4,5] Egill Snaebjörnsson Arnljots,[4,5] Maciej Godycki-Cwirko,[6] Anna Kowalczyk,[6] Tamara N Platteel,[2] Nicolaas P A Zuithoff,[2] Annelie A Monnier ,[1] Theo J M Verheij,[2] Cees M P M Hertogh,[1] Alma C van de Pol[2]

For numbered affiliations see end of article.

**Correspondence to**
Esther A R Hartman;
e.hartman1@amsterdamumc.nl

## ABSTRACT

**Introduction** Almost 60% of antibiotics in frail elderly are prescribed for alleged urinary tract infections (UTIs). A substantial part of this comprises prescriptions in case of non-specific symptoms or asymptomatic bacteriuria, for which the latest guidelines promote restrictiveness with antibiotics. We aim to reduce inappropriate antibiotic use for UTIs through an antibiotic stewardship intervention (ASI) that encourages to prescribe according to these guidelines. To develop an effective ASI, we first need a better understanding of the complex decision-making process concerning suspected UTIs in frail elderly. Moreover, the implementation approach requires tailoring to the heterogeneous elderly care setting.

**Methods and analysis** First, we conduct a qualitative study to explore factors contributing to antibiotic prescribing for UTIs in frail elderly, using semi-structured interviews with general practitioners, nursing staff, patients and informal caregivers. Next, we perform a pragmatic cluster randomised controlled trial in elderly care organisations. A multifaceted ASI is implemented in the intervention group; the control group receives care as usual. The ASI is centred around a decision tool that promotes restrictive antibiotic use, supported by a toolbox with educational materials. For the implementation, we use a modified participatory-action-research approach, guided by the results of the qualitative study. The primary outcome is the number of antibiotic prescriptions for suspected UTIs. We aim to recruit 34 clusters with in total 680 frail elderly residents ≥70 years. Data collection takes place during a 5-month baseline period and a 7-month follow-up period. Finally, we perform a process evaluation.

**Strengths and limitations of this study**

► The qualitative study allows for a comprehensive analysis of the factors at play in decision-making on urinary tract infections in frail elderly, which is essential to make progress in antibiotic stewardship in this setting.

► The pragmatic approach with its diverse international setting offers both broad applicability of results in general practice and elderly care medicine, and gives a chance to evaluate country-specific outcomes.

► The use of participatory action research (PAR) embedded within a cluster randomised trial is infrequent, and may offer valuable insights for future trials; however, a limitation of the tailored approach is that the results will not be exactly replicable.

► The process evaluation of the PAR approach will provide guidance for implementation in daily practice, including a toolbox with supportive educational materials.

► The COVID-19 pandemic began in the midst of the implementation process, undoubtedly affecting the process and results.

The study has been delayed for 6 months due to COVID-19 and is expected to end in July 2021.

**Ethics and dissemination** Ethical approvals and/or waivers were obtained from the ethical committees in Poland, the Netherlands, Norway and Sweden. The results will be disseminated through publication in peer-reviewed journals and conference presentations.

**Trial registration number** NCT03970356.

## INTRODUCTION

### Background and rationale

Suspected urinary tract infections (UTIs) account for the majority of antibiotic prescriptions in frail elderly. In recent years, consensus has been reached that non-specific symptoms in frail elderly are often not attributable to UTIs and do not require an antibiotic prescription.[1 2] However, it is estimated that between 32% and 62% of prescriptions for UTIs are inappropriately given to patients with only non-specific symptoms.[3 4] (Sundvall, NAPCRG conference 2017, unpublished) International efforts have been made to improve appropriate antibiotic prescribing: a decision tool to support physician's prescribing decisions was developed,[1] and recent guidelines promote restrictive antibiotic use for UTIs in frail elderly.[2] However, international evidence from a randomised controlled trial (RCT) on their efficacy in reducing inappropriate antibiotic use for UTIs is currently lacking.

Antibiotic prescribing decisions are known to be complex and influenced by many social and organisational factors.[5 6] In UTIs in frail elderly, this is further complicated by diagnostic uncertainties. First, frail elderly patients often present with non-specific symptoms. These symptoms should be evaluated for other causes but are often directly attributed to UTIs.[1 2 4 7 8] Second, interpretation of urinalysis is clouded by the high prevalence of asymptomatic bacteriuria, for which antibiotics are not needed.[2 7] A rigorous behavioural change is required from multiple healthcare professionals to improve antibiotic prescribing in this population. In order to develop effective antibiotic stewardship interventions (ASIs), it is essential to better understand the complex process leading to the decision to (not) prescribe antibiotics for alleged UTIs. Given the large variety in the organisation of elderly care, it is unlikely that a uniform ASI is effective.[9] Participatory action research (PAR) is a promising method that actively involves the healthcare professionals to implement an ASI tailored to their setting, while accounting for local barriers and facilitators.[10]

We set out to evaluate whether a multifaceted ASI is effective in reducing antibiotic prescribing for UTIs in frail elderly in various long-term care settings (in Poland, the Netherlands, Norway and Sweden). To accomplish the substantial behavioural changes that are needed, we believe we need a combination of qualitative methods for exploration and a PAR approach for implementation. First, we perform a qualitative study with semi-structured interviews to develop a conceptual model of factors contributing to antibiotic prescribing decisions in this population. Then, we conduct a cluster RCT in

frail elderly in care homes attended by general practitioners (GPs) using PAR to implement an ASI. Finally, we conduct a process evaluation.

### Objectives

- ► Obtain insights into all relevant factors that contribute to antibiotic prescribing for UTIs in frail elderly.
- ► Develop a conceptual model integrating these identified factors to guide the development of ASI for UTIs in frail elderly.
- ► Study the effects of the implementation of a multifaceted ASI on antibiotic prescription rates for UTIs in frail elderly.
- ► Evaluate the implementation process to understand the cluster RCT outcomes, and the added value of the PAR approach to implement ASIs.

## METHODS AND ANALYSIS

The Improving antibiotic Prescribing for UTIs in frail elderly (ImpresU) study consists of a qualitative study and a cluster RCT. Their integration is shown in figure 1.

### Qualitative study

The aims are to explore all relevant factors that contribute to antibiotic (non-)prescribing for UTIs in frail elderly, and to integrate these into a conceptual model to guide the development of effective ASIs.

#### Design and setting

An exploratory qualitative study using semi-structured interviews is conducted in Poland, the Netherlands, Norway and Sweden. Interviews are conducted with representatives of three relevant stakeholder groups in the setting of elderly care at home and in institutions: (1) GPs, (2) nursing staff and (3) patients and informal caregivers.

#### Eligibility criteria, recruitment and sample size

Recruitment takes place through the networks of the research teams per country. We use purposive sampling to reach variation within the representatives of each stakeholder group (eg, in setting, years of experience for healthcare professionals). All participants need to be capable and willing to provide informed consent and communicate personal thoughts in the local language. Patients need to be 70 years or older, and are not recruited during the acute phase of a disease. The aim is to conduct approximately 60 interviews (ie, 15 per country), preferably equally distributed over the three stakeholder groups.

#### Data collection and management

Topic lists and interview guides are designed based on literature and (clinical) experience from the researchers.[6] Pilot interviews are performed in each country to verify the appropriateness and completeness of the topic lists. All interviews are conducted in the native language and audio recorded. Basic demographic data (eg, gender, age) of participants are collected. Collected data and

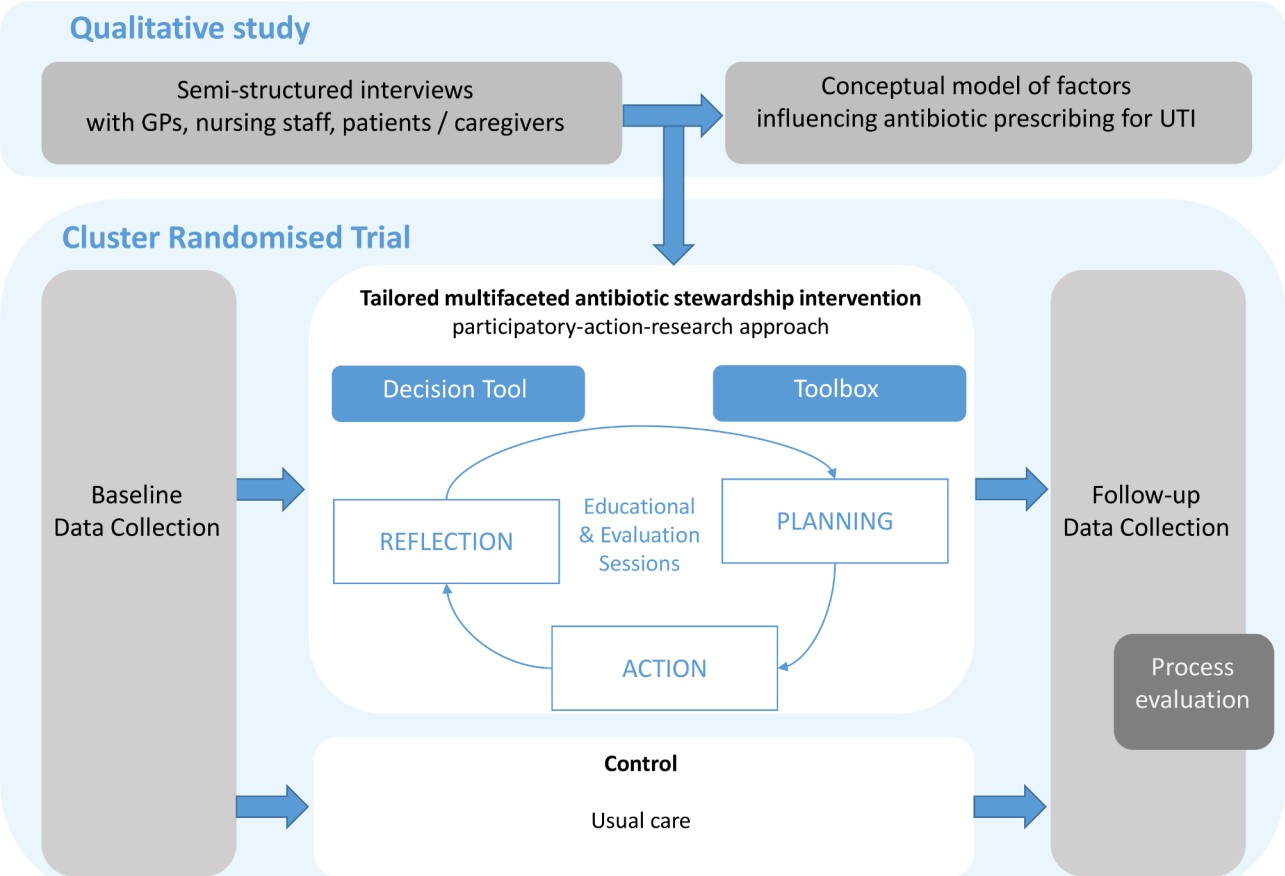

**Figure 1** Schematic overview of the interplay between the two studies. The qualitative study offers insights to tailor the antibiotic stewardship intervention in the cluster randomised controlled trial (RCT), through a country-specific local analysis. The cluster RCT consists of a baseline and follow-up period for data collection, with an intervention period or usual care in between (the timeline is provided in figure 3). A process evaluation follows at the end of the cluster RCT. GP, general practitioner; UTI, urinary tract infection.

transcripts are pseudonymised, using a code for each participant.

### Data analysis

Data are analysed with use of the framework method,[11] which consists of the following steps: (1) interviews are transcribed verbatim and translated into English; (2) the researchers (re) read the interviews for familiarisation; (3) two researchers independently code a first batch of interviews; (4) through consensus, a preliminary framework is formed; (5) the remaining interviews are coded using the framework; additions and changes are discussed within the research team; (6) data are organised in a framework matrix; (7) data are interpreted, and a conceptual model of factors is derived from the matrix.

### Cluster RCT

The trial aims to evaluate whether a decision tool for restrictive antibiotic use, implemented using a PAR approach, reduces antibiotic prescribing for UTIs in frail elderly. For this report, we used the Standard Protocol Items: Recommendations for Interventional Trials reporting guidelines.[12]

### Design and setting

A cluster RCT is performed in nursing homes in Poland, the Netherlands, Norway and Sweden, and in residential care homes and home care organisations in the Netherlands, attended by GPs. More details on the setting are provided in the online supplemental data 1. The cluster and unit of randomisation is the care organisation linked to the GP practice; one care organisation may be attended by multiple GP practices or vice versa. In the final months of the study period, a process evaluation is performed.

### Eligibility criteria and recruitment

Recruitment of clusters is performed through the networks of the research groups in Poland, the Netherlands, Norway and Sweden. The care organisations identify eligible patients, provide written study information, and ask whether they may be approached by the research team. Written informed consent from patients (or representatives in case of legal incapacity) is obtained by a visiting researcher or nurse.

For inclusion, patients need to be 70 years or older, have physical and/or mental disabilities and ADL dependency requiring care, do not use prophylactic antibiotics, do not

   

receive hospice care and are estimated not to have a very limited life expectancy (≤1 month). Patients are excluded when they start prophylactic antibiotics, start receiving hospice care, have a limited life expectancy (≤1 month), pass away, or move away from the cluster. Patients need to be included for at least 2 months to contribute data to the study.

## Sample size

The baseline incidence of UTI prescriptions is assumed to be 0.75 per patient year.[13–16] It has been shown that between 32% and 62% of these prescriptions are inappropriate, that is, not based on specific signs and symptoms.[3 4] (Sundvall NAPCRG conference 2017, unpublished) After implementation of the algorithm, we assume the prescription rate to be reduced from 0.75 to 0.4 prescriptions per person year. The intracluster correlation coefficient is expected to be 0.06, in line with related studies in the primary care and nursing home setting.[17 18]

For the sample size calculation, a Wilcoxon Test with an adjustment for cluster randomisation was performed. With an expected cluster size of 10 patients, each contributing 7 months in the follow-up period, one-sided testing, alpha of 0.05, and power of 0.8, it is estimated that 333 patients are needed, translating into a minimum of 34 clusters. To compensate for loss to follow-up, we assume 20 patients per cluster are needed. In sum, we aim to include 34 participating clusters, that is, 9 in each country, with in total 680 patients.

## Randomisation and blinding

Clusters are randomised to intervention or usual care, using SAS software V.9.4 by an independent data manager.[19] Block randomisation is used to assign clusters to intervention or control in each country, stratified on cluster size (small/medium/large). Due to the nature of the intervention, blinding is not possible; however, the aims of the study outcomes are not explicitly stated to the control clusters to avoid contamination.

## Intervention

The intervention clusters receive a multifaceted ASI. The control clusters provide care as usual. The intervention period was intended to last 4 months. After a month, it was interrupted by the first wave of the COVID-19 pandemic, resulting in a 6-month pause. On restart in September 2020, the pragmatic choice was made to restart the intervention period with a duration of 2–3 months, depending on the local situation.

### Decision tool and toolbox

At the core of the ASI is a decision tool to guide the use of antibiotics for suspected UTIs in frail elderly (online supplemental data 2).[1] It promotes an active monitoring approach in case only non-specific symptoms are present. This decision tool is incorporated in the Dutch UTI guideline for elderly care medicine and congruent with the Swedish and Norwegian UTI guidelines.[20–22] To support the implementation of the decision tool, a toolbox of educational materials is composed (figure 2 and online supplemental data 3). First, a generic toolbox is designed, centred around the decision tool. Next, it is tailored to become country-specific by the local researchers, based on the qualitative study data and any locally available materials. During the intervention period, further tailoring may take place within the participating cluster itself (figure 2).

### Implementation: modified PAR approach

The intervention is tailored based on an analysis of the interview data to identify country-specific barriers and facilitators. For example, the roles of the healthcare professionals and knowledge gaps in care for UTIs differ per country and need to be targeted accordingly. During the intervention period, the researchers and healthcare

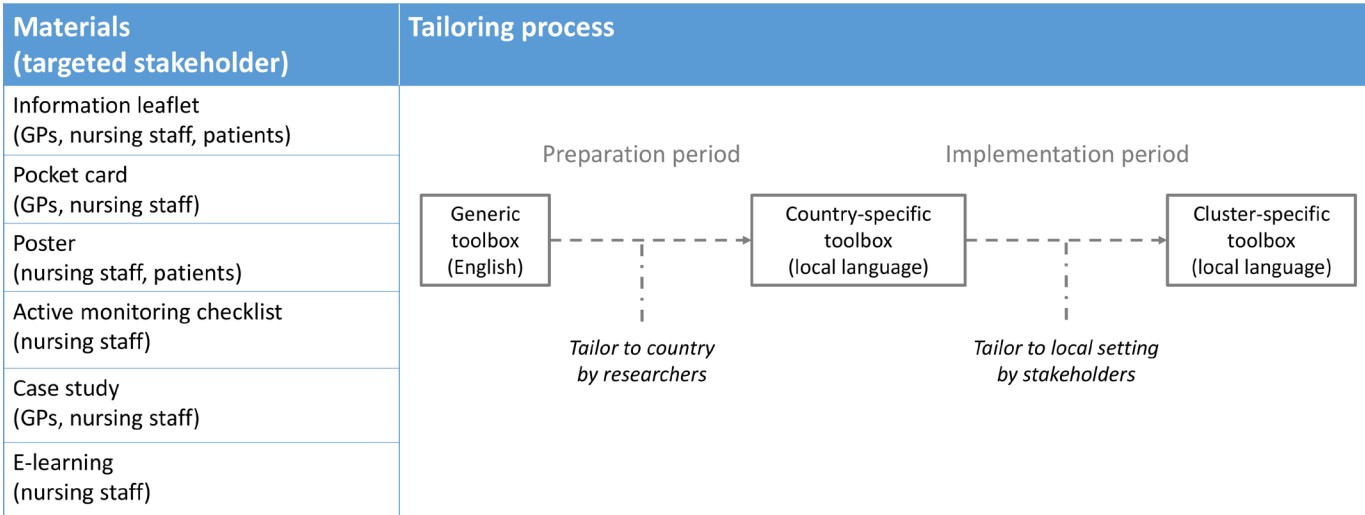

**Figure 2** Toolbox. The educational materials and targeted stakeholders in the generic toolbox are listed, and the tailoring process is shown. GP, general practitioner.

professionals together go through a cyclical process of reflection, planning and action during sessions for education and evaluation. These sessions combine a top-down and bottom-up approach; both education on the decision tool and any knowledge gaps identified in the qualitative study, as well as reflection and planning for local implementation. The aim is to go through at least two PAR cycles in each cluster, and to actively involve physicians as well as nursing staff. Further tailoring may be performed in each country and cluster locally.

## Outcome assessments
### Primary outcome measure
1. Number of prescriptions of antibiotics for suspected UTIs.

### Secondary outcome measures
2. Number of prescriptions of antibiotics for suspected UTIs in office hours.
3. Number of incorrect prescriptions of antibiotics for suspected UTIs.
4. Incidence of suspected UTIs.
5. Incidence of complications within 21 days after each UTI suspicion (presence yes/no of a complication: delirium, pyelonephritis, sepsis and renal failure).
6. Incidence of referral to a hospital within 21 days after each UTI suspicion.
7. Incidence of hospital admission within 21 days after each UTI suspicion.
8. Mortality.
9. Mortality within 21 days after each UTI suspicion.
   All outcomes are assessed during the follow-up period, and expressed per patient-year.

## Data collection
Data are collected during a 5-month baseline period and a 7-month follow-up period, through case report forms (CRFs) completed by the GP, nurse or researcher based on contact with a healthcare professional or medical file. The timeline for participating clusters and participants is displayed in figure 3.

For each participant, a CRF with patient characteristics is filled in at study entry consisting of items concerning demographics, ADL dependency measured through the Katz Index of Independence in Activities of Daily Living,[23] and relevant medical history. The GPs prospectively register each UTI suspicion on a short registration form, describing symptoms, diagnostics, and antibiotic treatment (primary and secondary outcomes). After 7 and 21 days, follow-up forms are filled in to assess the course of disease, any change in antibiotic treatment, complications, and mortality (primary and secondary outcomes). Overall mortality (secondary outcome) is registered on exclusion of a patient. Any missing data are retrospectively registered through consultation of GPs, nurses and/or access of the medical records.

Furthermore, anonymised data concerning COVID-19 incidence in the participating care organisations are registered during the follow-up period.

## Data management
Data are collected pseudonymised on paper forms, using a study code for each patient. Afterwards, they are electronically registered in the secured online database Research Online, according to regulations of the International Conference on Harmonisation - Good Clinical Practice.

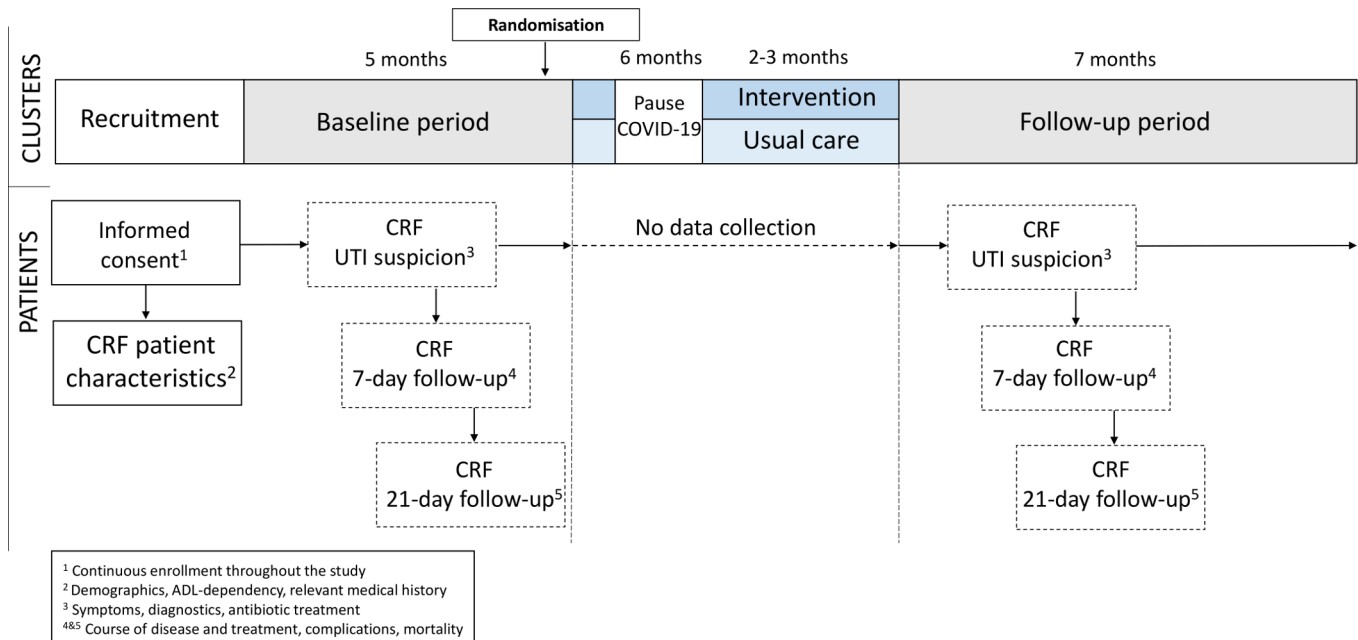

**Figure 3** Timeline of the cluster randomised controlled trial. The periods of data collection and procedures are shown for the clusters and participating patients. CRF, case report form; UTI, urinary tract infection.

Research Online has multiple validation rules built into the eCRFs. The data cleaning process is supported by automatically and manually generated queries. At the end of the study, all data will be locked. Dedicated data sets are provided to the researchers for analysis. Data are kept securely for at least 15 years.

## Data analysis

The analysis will follow the intention-to-treat principle. For the primary outcome, a generalised linear mixed model for Poisson distributions will be used. In case the assumptions for Poisson distributions are insufficiently met, other distributions will be considered (ie, negative binomial, generalised Poisson, zero-inflated Poisson). A random intercept will be included to correct for clustering within care facility and/or GP, and an additional random intercept will be included to correct for repeated measurements in patients. When results indicate no or very low clustering at the facility/GP or patient level, the corresponding random intercept will be excluded from the analysis. The comparison between intervention and control group, estimated with the time by treatment interaction, will be reported as Rate Ratio's with a 95% CI and a corresponding p value. In a second model, prespecified prognostic factors will be added: age, gender, ADL dependency, presence of an indwelling catheter, dementia, recurrent UTIs, diabetes mellitus, and kidney disorders. In case there are missing values on baseline variables that were selected as potential confounders, multiple imputation will be considered. Furthermore, subgroup analysis will be performed to assess outcomes in groups per country, with different gender, age, presence of dementia, urinary incontinence and indwelling catheter.

## Process evaluation

A process evaluation is conducted in the care organisations participating in the cluster RCT. The framework described by Saunders et al[24] is used. Elements that are assessed include fidelity, dose delivered/received, reach, recruitment and context (including COVID-19 impact). Data are collected through documentation of the intervention process by the researchers, and through questionnaires with closed-ended and open-ended questions to participating healthcare personnel. Quantitative data will be reported using descriptive statistics; thematic analysis will be performed on the qualitative data.

## Patient and public involvement

In the qualitative study, patients and informal caregivers are interviewed. These data were taken into account in the intervention implementation in the cluster RCT. In the process of the design of the cluster RCT, a meeting was held with representatives of Network Utrecht, care for the elderly (NUZO), Julius Centre, University Medical Centre Utrecht, The Netherlands. Their suggestions on the protocol were taken into account; for example, on patient-directed toolbox materials.

## DISCUSSION

We perform a European qualitative study exploring factors influencing decision making on UTIs in frail elderly, and a pragmatic cluster RCT to assess the effect of a decision tool to improve antibiotic prescribing for UTIs in frail elderly, implemented using a PAR approach. We believe this combination of methodologies is essential to address the complexity of decision-making on UTIs in this population. Drawing lessons from the Improving Rational Prescribing of Antibiotics in Long-term Care Facilities (IMPACT) study,[25] we are the first to apply this in a diverse international setting.

The PAR approach for implementation allows us to embrace the heterogeneity of the elderly care settings within and between countries.[26] With large-scale nursing homes in some countries and small-scale living facilities in others, an identical ASI for each healthcare professional will not be effective. Tailoring the intervention using PAR promotes bottom-up engagement of healthcare professionals, thereby enabling the required behavioural changes for lasting effects.

Inherent to the tailored approach are limits in the ability to exactly replicate our results. Nevertheless, the methods are replicable, and we believe our results will be widely applicable. The qualitative study will offer in-depth understanding of the factors involved in decisions on UTI, thereby creating opportunities for future ASI development. Our robust trial design, in line with epidemiological recommendations for evaluating ASI,[27] will provide evidence on the application of the latest UTI guidelines. Furthermore, our process evaluation will generate understanding on the ASI and its components in the various settings, and will provide lessons on the use of PAR in future trials. A practical implementation package will become available, with relevant toolbox materials and lessons for daily practice to be tailored to any setting. A further limitation of our study is that we cannot collect data on overall antibiotic use, as we focus on prospective registration in included patients of suspected UTIs only.

The cluster RCT was interrupted by the first wave of the COVID-19 pandemic during the intervention period, and was forced to pause for 6 months. Restarting required much flexibility from the participating care organisations, where patient care already suffered from the pandemic. Sessions for the intervention meeting had to be repeated (mostly online). Furthermore, the 6-month delay and further COVID-19 waves regrettably continue to lead to the passing away of participants, increasing the need for new recruitment. As randomisation takes place per country, we presume effects of COVID-19 on our population characteristics and outcomes, if any, will be balanced between intervention and control clusters.

In conclusion, we aim to evaluate the effectiveness of a multifaceted ASI to reduce antibiotic prescribing for UTIs in frail elderly through a qualitative study and cluster RCT in Poland, the Netherlands, Norway and Sweden. Our tailored approach within the diverse setting

is promising to yield broadly applicable results, even if currently challenged by the COVID-19 pandemic.

## ETHICS AND DISSEMINATION
### Participant safety and monitoring
The cluster RCT is considered low risk, as the intervention corresponds to current guidelines. There is no data monitoring committee, and any SAEs are not reported. No interim analyses are planned. For both the qualitative study and cluster RCT respectively, ethical approval was given by the Committee of Bioethics of the Medical University of Lodz, Poland (RNN/381/18/KE and RNN/260/19/KE), the Regional Committee for Medical and Health Research Ethics in Norway (2018/2191/ REK sør-øst A and 2018/2521/REK sør-øst A), and the Swedish Ethical Review Authority (2019-00504 and 2019-00796/1228-18 (2019-02541)). In the Netherlands, the Medical Ethics Review Committees established that approval was not required since the Medical Research Involving Human Subjects Act does not apply (2018.500 VU University Medical Centre and WAG/mb/19/012207 University Medical Centre Utrecht). Substantial protocol modifications are communicated to ethical committees and the trial register. Dissemination will take place through publication and presentations. Furthermore, an implementation package will be developed.

### Trial status
Currently, the cluster RCT is ongoing and expected to finish in July 2021. Database lock will take place in September 2021.

**Author affiliations**
[1]Department of Medicine for Older People, Amsterdam Public Health Research Institute, Amsterdam UMC, Vrije Universiteit Amsterdam, Amsterdam, The Netherlands
[2]Julius Center for Health Sciences and Primary Care, University Medical Center Utrecht, Utrecht University, Utrecht, The Netherlands
[3]The Antibiotic Centre for Primary Care, Department of General Practice, Institute of Health and Society, University of Oslo, Oslo, Norway
[4]General Practice/Family Medicine, School of Public Health and Community Medicine, Institute of Medicine, Sahlgrenska Academy, University of Gothenburg, Gothenburg, Sweden
[5]Research, Education, Development & Innovation, Primary Health Care, Region Västra Götaland, Sweden
[6]Centre for Family and Community Medicine, Faculty of Health Sciences, Medical University of Lodz, Lodz, Poland

**Acknowledgements** We wish to thank Sofia Sundvall and Sara Sofia Lithén, research nurses, for their ongoing efforts in data collection. Furthermore, we would like to express our gratitude to the participating general practices and elderly care organisations for their prolonged contributions despite the current pandemic.

**Contributors** CMPMH, TJMV, ML, P-DS and MG-C conceptualised the study and obtained funding. For the qualitative study, AAM drafted the protocol with EARH, WGG, SRH-O, ML, SH, P-DS, IS, ESA, AK, MG-C and CMPMH. For the cluster RCT, ACvdP drafted the protocol with EARH, WGG, SRH-O, ML, SH, P-DS, RG, ESA, MG-C, AK, TNP, NPAZ, TJMV and CMPMH. NPAZ wrote the statistical analysis plan with EARH and ACvdP. EARH, WGG, ACvdP, AAM, TNP, SRH-O, SH, AK, ESA and P-DS designed the process evaluation. The manuscript was drafted by EARH and critically revised by all authors. All authors read and approved the final manuscript.

**Funding** This work was supported by JPI AMR with reference number JPIAMR_2017_P007, through national funding agencies: National Science Centre Poland (UMO-2017/25/Z/NZ7/03024), ZonMw The Netherlands(549003002), the Research Council of Norway (284253/H10), and The Swedish Research Council (2017-05975). The Healthcare Board, Region Västra Götaland (N/A) partially funded the Swedish part of the study.

**Disclaimer** The funders have no role in or authority on study design, data collection, management, analysis or interpretation, writing and submission of reports for publication.

**Competing interests** None declared.

**Patient consent for publication** Not applicable.

**Provenance and peer review** Not commissioned; externally peer reviewed.

**ORCID iDs**
Esther A R Hartman http://orcid.org/0000-0003-0907-2476
Annelie A Monnier http://orcid.org/0000-0002-5844-286X

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
