## [Reviewer comments · BMJ Open]

ARTICLE DETAILS

TITLE (PROVISIONAL)	A multifaceted antibiotic stewardship intervention using a participatory-action-research approach to improve antibiotic prescribing for urinary tract infections in frail elderly (ImpresU): study protocol for a European qualitative study followed by a pragmatic cluster randomised controlled trial.
AUTHORS	Hartman, Esther; Groen, Wim; Heltveit-Olsen, Silje; Lindbaek, Morten; Høy, Sigurd; Sundvall, Pär-Daniel; Gunnarsson, Ronny; Skoglund, Ingmarie; Snaebjörnsson Arnljots, Egill; Godycki-Cwirko, Maciej; Kowalczyk, Anna; Platteel, Tamara; Zuithoff, Nicolaas; Monnier, Annelie; Verheij, Theo; Hertogh, Cees; van de Pol, Alma

VERSION 1 – REVIEW

REVIEWER	Olaru, Ioana D. LSHTM
REVIEW RETURNED	14-Jun-2021

GENERAL COMMENTS	This is the protocol of a study aiming to improve antibiotic prescribing in frail elderly. The study is important given the considerable over-prescription of antibiotics in this patient population. The study will be conducted in four countries in Europe and will comprise two main components: a cluster randomized controlled trial evaluating a stewardship intervention and qualitative interviews with relevant stakeholders and patients that will inform the intervention. Comment - Methods section: Please provide more detail on how the relevant stakeholders are selected for the interviews. How will the researchers ensure that the selected participants are representative? For the healthcare providers – from how many healthcare facilities will they be selected?
---

REVIEWER	Mun, SEok Inje University College of Medicine, Division of Infectious Diseases
REVIEW RETURNED	17-Jun-2021

GENERAL COMMENTS	I think that this is a meaningful trial for reducing misuse of antibiotics and antimicrobial resistance. The aim of the study group is whether the use of the decision tool and the PAR approach can improve antibiotic prescribing for urinary tract infections (UTIs) in frail elderly. They conducted a qualitative study and then proceeded to a cluster randomized controlled trial. They aimed to include 680 patients in the cluster randomized controlled
--

trial, considering loss to follow-up. Clusters were randomized by an independent data manager. Block randomization was used to assign clusters to intervention or control in each country, stratified on cluster size. Blinding was not possible. The primary outcome was the number of antibiotic prescriptions for suspected UTIs. A limitation of the PAR approach is that the results will not be exactly replicable.

In the decision tool, symptoms and signs are mainly used for diagnosing UTIs. However, symptoms and signs may be nonspecific in frail elderly. Furthermore, because blinding was not possible, the intention of the participants could influence the outcomes. It may need to evaluate the number of antibiotic prescriptions for other causes in addition to UTIs.

The order of Figure 2 and 3 appears to be reversed.

VERSION 1 – AUTHOR RESPONSE

Response to comment by Reviewer 1

Dr. Ioana D. Olaru, LSHTM, Biomedical Research and Training Institute

Point 2: This is the protocol of a study aiming to improve antibiotic prescribing in frail elderly. The study is important given the considerable over-prescription of antibiotics in this patient population. The study will be conducted in four countries in Europe and will comprise two main components: a cluster randomized controlled trial evaluating a stewardship intervention and qualitative interviews with relevant stakeholders and patients that will inform the intervention.

Response 2: We thank the reviewer for the time, elaborate notes, and acknowledgement of the importance of our study. No changes were made in response to this statement.

Point 3: Comment - Methods section: Please provide more detail on how the relevant stakeholders are selected for the interviews. How will the researchers ensure that the selected participants are representative? For the healthcare providers – from how many healthcare facilities will they be selected?

Response 3: We agree with the reviewer that the recruitment process requires a more elaborate description. We aim to explore all relevant factors that contribute to antibiotic prescribing for UTIs in frail elderly. We aspire for results representative for the setting of elderly care through interviewing a wide variety of stakeholders. We use purposive sampling to reach this variety, e.g. in years of experience, educational level (nurse assistant or nurse), and setting (home care, nursing home). We aim to conduct 60 interviews, of which 20 interviews with patients/caregivers and 40 with health care professionals. Health care professionals will be invited through many different healthcare facilities to represent various settings (GP practices, nursing homes, residential care homes, and home care organizations) in four countries. In the manuscript, we have provided more details in the section on recruitment within the Methods section on page 5.

Response to comment by Reviewer 2

Dr. SEok Mun, Inje University College of Medicine

Point 4: I think that this is a meaningful trial for reducing misuse of antibiotics and antimicrobial resistance. The aim of the study group is whether the use of the decision tool and the PAR approach can improve antibiotic prescribing for urinary tract infections (UTIs) in frail elderly. They conducted a qualitative study and then proceeded to a cluster randomized controlled trial. They aimed to include 680 patients in the cluster randomized controlled trial, considering loss to follow-up. Clusters were randomized by an independent data manager. Block randomization was used to assign clusters to intervention or control in each country, stratified on cluster size. Blinding was not possible. The primary outcome was the number of antibiotic prescriptions for suspected UTIs. A limitation of the PAR approach is that the results will not be exactly replicable.

Response 4: We thank the reviewer for the time, elaborate assessment, and positive feedback while understanding the limitations. We discuss the limitations of our PAR approach in the section on “Strengths and limitations” and in the discussion. The results will indeed be not exactly replicable. However, our design is replicable, and the PAR approach allows for the necessary tailoring of the intervention to the heterogeneous setting. No changes were made in the manuscript in response to this statement.

Point 5: In the decision tool, symptoms and signs are mainly used for diagnosing UTIs. However, symptoms and signs may be nonspecific in frail elderly.

Response 5: We thank the reviewer for bringing up this item. The reviewer here touches upon a crucial point in the rationale for our study. Indeed, many frail elderly present with non-specific symptoms. According to recent guidelines and the decision tool, antibiotics are not indicated for most of these patients. We have made adjustments in the first paragraph of the Introduction to further clarify the role of non-specific symptoms.

Point 6: Furthermore, because blinding was not possible, the intention of the participants could influence the outcomes. It may need to evaluate the number of antibiotic prescriptions for other causes in addition to UTIs.

Response 6: For us, it is not entirely clear what the reviewer indicates as possible consequences of the fact that blinding is not possible in our study. The aim of our intervention is to educate health care professionals and increase their awareness on appropriate antibiotic use: we thus indeed wish for their intention to influence the outcomes; namely reduce unnecessary antibiotic prescriptions for UTIs.

The reviewer suggests to evaluate antibiotic prescriptions for other indications than UTIs. In our study design, we prospectively follow patients and data are collected when they have a suspected UTI. The strength of this prospective design includes the possibility to have detailed information on suspected UTIs and their course of disease. Our CRFs include the possibility for GPs to indicate whether an alternative cause for the symptoms is suspected. However, a limitation is that we cannot evaluate overall antibiotic use, as no data are collected if there is no UTI suspicion at all. We added this limitation to the Discussion section.

Point 7: The order of Figure 2 and 3 appears to be reversed.

Response 7: We thank the reviewer for the correction, and we have now uploaded the files with the correct names.